# Olive Mill Waste-Water Extract Enriched in Hydroxytyrosol and Tyrosol Modulates Host–Pathogen Interaction in IPEC-J2 Cells

**DOI:** 10.3390/ani14040564

**Published:** 2024-02-07

**Authors:** Flavia Ferlisi, Chiara Grazia De Ciucis, Massimo Trabalza-Marinucci, Floriana Fruscione, Samanta Mecocci, Giulia Franzoni, Susanna Zinellu, Roberta Galarini, Elisabetta Razzuoli, Katia Cappelli

**Affiliations:** 1Department of Veterinary Medicine, University of Perugia, 01623 Perugia, Italy; flavia.ferlisi@studenti.unipg.it (F.F.); samanta.mecocci@unipg.it (S.M.); katia.cappelli@unipg.it (K.C.); 2National Reference Center of Veterinary and Comparative Oncology (CEROVEC), Istituto Zooprofilattico Sperimentale del Piemonte, Liguria e Valle d’Aosta, 16129 Genova, Italy; chiaragrazia.deciucis@izsto.it (C.G.D.C.); floriana.fruscione@izsto.it (F.F.); elisabetta.razzuoli@izsto.it (E.R.); 3Department of Animal Health, Istituto Zooprofilattico Sperimentale della Sardegna, 07100 Sassari, Italy; giulia.franzoni@izs-sardegna.it (G.F.); susanna.zinellu@izs-sardegna.it (S.Z.); 4Centro Specialistico Sviluppo Metodi Analitici, Istituto Zooprofilattico Sperimentale dell’Umbria e delle Marche “Togo Rosati”, 06126 Perugia, Italy; r.galarini@izsum.it

**Keywords:** polyphenols, IPEC-J2, cytokine, defensin, immunomodulation, *Salmonella* spp.

## Abstract

**Simple Summary:**

Olive mill waste-water (OMWW) is a liquid waste produced by the olive oil industry that has been recently regarded as a good source of polyphenols. Phenolic molecules are among the most active secondary molecules in the gut for their antioxidant, anti-inflammatory and antimicrobial effects. They may also contribute to positively changing the distribution of gut microbial species, but their effects have not been widely explored in pigs. The intestinal porcine epithelial cell line IPEC-J2 represents a good model for the study of innate immunity and inflammatory response in animal intestinal diseases and has already been used to investigate the effect of phytogenic feed additives on swine intestinal epithelium. This study aimed to evaluate the *in vitro* effects of an OMWW extract enriched in polyphenols on *Salmonella typhimurium* (*S. typhimurium*) infection in IPEC-J2 cells. Polyphenols extracted from OMWW showed the ability to regulate the host–pathogen interaction by decreasing *S. typhimurium* invasiveness and modulating the expression of many innate immune genes.

**Abstract:**

The dietary supplementation of olive oil by-products, including olive mill waste-water (OMWW) in animal diets, is a novel application that allows for their re-utilization and recycling and could potentially decrease the use of antibiotics, antimicrobial resistance risk in livestock species, and the occurrence of intestinal diseases. *Salmonella serovar typhimurium* is one of the most widespread intestinal pathogens in the world, causing enterocolitis in pigs. The aim of this study was to investigate the effect of an OMWW extract enriched in polyphenols (hydroxytyrosol and tyrosol) in the immune response of an intestinal porcine epithelial cell line (IPEC-J2) following *S. typhimurium* infection. Cells were pre-treated with OMWW-extract polyphenols (OMWW-EP, 0.35 and 1.4 µg) for 24 h and then infected with *S. typhimurium* for 1 h. We evaluated bacterial invasiveness and assayed IPEC-J2 gene expression with RT-qPCR and cytokine release with an ELISA test. The obtained results showed that OMWW-EP (1.4 µg) significantly reduced *S. typhimurium* invasiveness; 0.35 µg decreased the IPEC-J2 gene expression of *IL1B*, *MYD88*, *DEFB1* and *DEFB4A*, while 1.4 µg down-regulated *IL1B* and *DEFB4A* and increased *TGFB1.* The cytokine content was unchanged in infected cells. This is the first study demonstrating the *in vitro* immunomodulatory and antimicrobial activity of OMWW extracts enriched in polyphenols, suggesting a protective role of OMWW polyphenols on the pig intestine and their potential application as feed supplements in farm animals such as pigs.

## 1. Introduction

The extraction of olive oil produces a series of by-products, including olive mill waste-water (OMWW)—olive vegetation water diluted in the water used during the oil-extraction process. This by-product is characterized by a high organic material load, ranging from 36.07 g/L to 230 g/L, and a content of phenolic compounds that varies from 0.9 to 30.5 g/L [1,2]. The large amount of this by-product (30 million m^3^), produced every year in the Mediterranean basin, contributes to environmental pollution due to the high presence of organic compounds, including phenolic ones [3,4,5,6]. This by-product comprises about 50% of the total phenolic compounds of the olive fruit [7], with different phenolic types, mainly tyrosol, hydroxytyrosol, verbascoside and oleuropein [8,9,10], which are highly known for their antioxidant, antimicrobial and anti-inflammatory activities [11,12,13,14,15]. The supplementation of olive by-products, including OMWW, as a source of polyphenols in animal diets potentially represents an innovative strategy for olive oil waste recycling, in line with the current concept of the circular economy [16,17,18,19,20,21,22].

In the swine industry, the use of antibiotics can favor the occurrence of antimicrobial resistance in bacteria of the pig intestinal microbiome, therefore increasing the risk of severe intestinal diseases and impairing the pig’s growth performance, especially at the weaning stage [23,24,25]. For this reason, this habit has been limited in various countries. Among pig intestinal diseases, salmonellosis is one of the most common and it represents a severe problem for the swine industry worldwide [26]. *Salmonella enterica* serovar *typhimurium* (*S. typhimurium*) is the agent of a very widespread enterocolitis form, which can be subclinical, but it can also be associated with a reduction in both productive performance and average daily gain in pigs [27,28].

In order to restrict the use of antibiotics, novel feeding strategies are required to modulate intestinal and immunological functions, as well as to improve the development and health of the swine gastrointestinal tract [23]. Given the correlation between bioactive molecules, such as polyphenols, and the pig intestinal microbiota and immune response to enteric diseases, their use can have a good impact on pig gut health [23]. The health benefits of polyphenols derive from their antioxidant, anti-inflammatory, and/or gene-regulating effects in tissues. Several studies showed that they help decrease the risk of many diseases, including intestinal ones, but the mechanisms correlated are not clear and need further investigation [29,30,31]. At present, they can be considered among the most active secondary bioactive molecules in the gut, contributing to beneficial changes in the distribution of gut microbial species, reducing pathogenic bacteria, and/or promoting the growth of probiotics [29,32,33]. A number of *in vivo* studies demonstrated that the administration of dietary polyphenols resulted in a reduction of pathogenic species and an increase in probiotic species in the intestinal microbiota of rats, pigs, and calves [32,34,35,36,37]. Olive oil by-products rich in polyphenols (e.g., olive leaf extract) were able to interfere with the growth of intestinal bacteria, including *Salmonella* [38,39]. Compared to human and laboratory animals (e.g., rats and mice), responses to polyphenols have been less explored in farm animals, such as pigs [29]. However, it has been recently demonstrated that the supplementation of natural polyphenols in piglets could contribute to alleviating weaning stress and improve intestinal barrier function, thus providing a nutritional strategy to protect intestinal health [40,41]. Other studies examined changes in the pig gut microbiome after the consumption of plant polyphenols, thanks to their ability to reduce oxidative stress and inflammation [42,43] and modulate immune cells and gut microbiota composition [29,44,45,46]. This action contributes to an improvement in intestinal bacterial function, decreases the release of microbial components into the circulation, and stimulates host immune response [47].

A suitable *in vitro* model to assess the immunomodulatory properties of polyphenols is represented by the porcine jejunal epithelial cell line IPEC-J2. This continuous cell line provides a valuable model to study both innate immunity and inflammatory responses in human and animal intestinal diseases [26,48,49,50,51,52]. Indeed, IPEC-J2 cells are intestinal porcine enterocytes isolated from the jejunum of an unsuckled neonatal pig, which showed the ability to express and produce cytokines, toll-like receptors (TLRs), defensins, and mucins [53]. In particular, these cells spontaneously secrete the pro-inflammatory chemokine IL-8 and possess ideal characteristics for *in vitro* studies on host–intestinal pathogen interactions [49,50,54]. Indeed, the primary host-cell barrier against pathogens is represented by the mucosal innate immune system, which is characterized by toll-like receptor (TLR) pathways, NF-kB signaling activation (with inflammatory cytokine release), and Type-I Interferon (IFN) responses [49,55]. Moreover, gastrointestinal tract homeostasis can be maintained when the immune response against commensal bacteria is controlled. When this equilibrium is compromised, excessive immune response causes an inflammatory condition [56]. Besides epithelial cells’ mechanical function, their role in gut microbiota homeostasis was recently recognized, as they are involved in maintaining the balance between host microbial components and gut immune cells [57]. In addition, IPEC-J2 cells mime the physiological characteristics of intestinal cells and have therefore been employed in several studies on *Salmonella* infections [53,58], providing valuable information on host responses to this bacteria. In fact, invasion with *S. typhimurium* in IPEC-J2 cells was comparable to that occurring in porcine ileal mucosal explants [59]. This cell line has been employed in studies focused on pigs’ innate immune response to dietary treatments [60], which can be regarded as a reference for *in vitro* studies of innate immunity in neonatal intra-epithelial cells after dietary stimuli [48,60]. These cells showed high morphological and functional similarities to porcine enterocytes *in vivo*; therefore, they were employed to evaluate the effects of phytogenic feed additives on swine intestinal epithelium [61]. In recent years, various plant-feed additives have demonstrated antioxidant, antimicrobial, and anti-inflammatory actions and other supporting barrier functions in IPEC-J2 cells [62,63,64,65].

With this study, we aimed to investigate *in vitro* the IPEC-J2 response to *S. typhimurium* infection after a pre-treatment with OMWW-extract polyphenols (OMWW-EP), to evaluate the influence on bacterial invasion and immune cells’ gene expression.

## 2. Materials and Methods

### 2.1. Olive Mill Waste-Water Extract and Composition

The OMWW extract enriched in polyphenols (hydroxytyrosol and tyrosol) was provided by Stymon Natural Products P.C., Patras, Greece (www.stymon.com, accessed on 21 December 2023). This product derives from OMWW of the olive (*Olea Europaea* L.) variety Koroneiki and is produced based on a unique patent (Patent number 1,010,150 IOBE (INT.CL.2021.01) A23L 19/00 A23L 33/105, Stymonphen Liquid) using only green technologies. Its total polyphenol content was equal to 15,000 ± 592 mg/kg, according to the Folin–Ciocalteu method [66]; hydroxytyrosol and tyrosol were the main phenolic compounds (8784 mg/kg and 1638 mg/kg, respectively), detected by HPLC-DAD [67]. The stock solution was filtered, vortexed, and diluted in phosphate-buffered saline (PBS, Euroclone, Milan, Italy) to reach 1400 µg/mL; from this, different scalar concentrations of polyphenols (0.35; 0.7; 7; 14; 70; 140 µg) were obtained for the successive analyses by diluting them in complete culture medium.

### 2.2. Cell Cultures

Porcine jejunal epithelial cells (IPEC-J2, IZSLER Cell Bank code BS CL 205) were grown in a mixture (1:1) of Dulbecco’s Modified Eagle (DMEM) (Euroclone, Milan, Italy) and Nutrient Mixture F-12 (F12) (Euroclone, Milan, Italy) enriched with 10% Fetal Bovine Serum (FBS, GIBCO^TM^, Thermofisher Scientific, Milan, Italy), 1% L-glutamine solution (Euroclone, Milan, Italy) and 1% penicillin/streptomycin solution (Euroclone, Milan, Italy) and kept in culture at 37 °C under 5% CO_2._

#### 2.2.1. Cell Viability

First, to determine the most suitable amount of OMWW extract to be used on IPEC-J2 cells, we tested different scalar phenolic dosages using a 2,3-bis-(2-methoxy-4-nitro-5-sulphophenyl)-2H-tetrazolium-5-carboxanilide (XTT) assay, according to the manufacturer’s instructions (XTT Cell Viability Assay, Cell Signaling Technology, Milan, Italy). In brief, IPEC-J2 cells were plated on 96-well plates (100 µL per well, 0.1 × 10^5^) in complete culture medium and incubated for 24 h at 37 °C under 5% CO_2_ until confluence. The day after, the seeding cells on the 96-well plates were exposed to different doses of OMWW-EP (0.35, 0.7, 1.4, 14, 70, 140 µg), and untreated cells were employed as a negative control. Two independent experiments were performed, each including four technical replicates (four wells) for each of the seven experimental conditions: untreated cells (control) and cells treated with six different doses of OMWW-EP (0.35, 0.7, 1.4, 14, 70, and 140 µg). An XTT assay was performed at 24 h and at the end of the treatments, and the cell culture medium was removed and replaced with 100 µL of the fresh DMEM/F12 medium supplemented with XTT detected solution (1:50). The plates were then incubated again at 37 °C under 5% CO_2_ for 2 h, and the absorbance was measured at 450 nm using a multimode microplate reader (Glomax, Promega, Milan, Italy). This assay was performed two times for each phenolic concentration.

#### 2.2.2. Bacterial Invasion

An isolate of *S. typhimurium* strain (ATCC 14028) was used to evaluate bacterial invasion in IPEC-J2 cells. In three independent experiments, IPEC-J2 cells were seeded into a 12-well plate (1 mL per well, 1.5 × 10^5^ cells/mL) and incubated until confluence. Cells were treated with OMWW-EP (0.35 μg and 1.4 µg) for 24 h. *S. typhimurium* was stored at −80 °C until use, then thawed and grown overnight (18–24 h at 37 °C) in Brain Heart Infusion (BHI) (Sigma, Saint Louis, MO, USA). Then, it was sub-cultured in BHI and incubated for 2 h at 37 ± 1 °C to obtain a mid-log phase culture. The strain was pelleted and re-suspended in DMEM/F12 and L-glutamine medium to obtain a concentration of 10^8^ CFU/mL and used to infect pig intestinal IPEC-J2 cells pre-treated with 0.35 μg and 1.4 µg of OMWW-EP for 24 h; infected cells without polyphenolic pre-treatment were used as comparison, while uninfected cells were employed as a negative control. For each of the three independent experiments, one plate was used, employing one well for each replicate, resulting in four replicates for each experimental condition: cells without phenolic pre-treatment and infected (ST); cells pre-treated with two OMWW-EP dosages and infected (ST + 0.35 µg POL; ST + 1.4 µg POL). Cells were stimulated with 1 mL/well of bacterial suspension at 10^8^ CFU/mL and incubated at 37 °C under 5% CO_2_ for 1 h. Then, monolayers were washed five times with DMEM/F12 and L-glutamine medium (1 mL/well) and treated with 1 mL of colistin 300 μg/mL at 37 °C under 5% CO_2_ for 2 h to remove all extracellular bacteria. Cells were washed four times with medium and lysed by adding 200 μL/well 1% of Triton X-100 (Merck KgaA, Darmstadt, Germany) in PBS (Euroclone, Milan, Italy) at room temperature for 5 min (min); then, they were blocked by adding 800 μL of PBS to each well. The resulting cell suspension was vortexed, serially diluted in PBS, and seeded on XLD (Sigma, Saint Louis, MO, USA); then, it was incubated for 24–48 h at 37 °C for intracellular bacterial counts.

#### 2.2.3. Modulation of the Immune Response

Cells from the IPEC-J2 line were seeded into 12-well plates (1 mL per well, 3 × 10^5^ cells/mL) and then incubated at 37 °C under 5% CO_2_ until confluence. Two experimental designs were applied: the first one to evaluate the effect of OMWW-EP on IPEC-J2 gene expression and cytokine release, and the second one to investigate cellular pathways modulated by OMWW pre-treatment behind host–pathogen interactions in *S. typhimurium* infection. For the first experiment, cells were treated with OMWW-EP at 0.35 µg, 0.7 µg, 1.4 µg or 7 µg for 24 h, alongside untreated controls. A total of three independent experiments comprising two replicates each (one well for each replicate) were used for each experimental condition: cells with medium only (control); cells treated with OMWW-EP (0.35 µg POL, 0.7 µg POL, 1.4 µg POL and 7 µg POL). For the second, cells were treated with OMWW-EP at 0.35 µg or 1.4 µg for 24 h, then infected with 1 mL of 10^8^ CFU/mL *S. typhimurium* suspension and incubated at 37 °C under 5% CO_2_ for 1 h. Moreover, cells infected with *S. typhimurium* without OMWW-EP pre-treatment were used as a control of the infection, and cells with the medium only were used as an untreated and uninfected control. To summarize, four independent experiments, including three replicates (one well for each replicate), were used for each experimental condition: cells with medium only (control); infected cells only (ST); cells pre-treated with polyphenols (0.35 and 1.4 µg); and infected (0.35 µg POL + ST; 1.4 µg POL + ST). After the first incubation, cells were washed five times and again incubated in their medium at 37 °C under 5% CO_2_ for 3 h. The resulting IPEC-J2 cell supernatants were stored at −80 °C until our evaluation of the cytokine contents. In parallel, cells were lysed with 400 µL of RLT Buffer (Qiagen, Hilden, Germany) and, after incubation for 10 min (min) at room temperature, collected and stored at −80 °C until RNA extraction and RT-q PCR analysis.

### 2.3. RNA Extraction and Reverse Transcription Quantitative PCR (RT-qPCR)

Total RNA was extracted from the cells described in Section 2.2.3 for both the experimental designs using Rneasy Mini Kit (Qiagen s.r.l., Milan, Italy) in the Qiacube System (Qiagen s.r.l., Milan, Italy), in accordance with the manufacturer’s instructions. The quality of extraction was assessed using a Qubit 3.0 Fluorometer (Thermo Fisher Scientific, Waltham, MA, USA). The same amount of RNA for each sample (250 ng) was reverse-transcribed into cDNA using a iScript cDNA Synthesis Kit (Bio-Rad, Milan, Italy). Amplification was performed on a CFX96^TM^ Real-Time System (Bio-Rad, Milan, Italy) using SoFast^TM^ Eva Green Supermix (Bio-Rad, Milan, Italy) following a protocol previously described [48]. Primers of target genes, coding for C-X-C motif chemokine ligand 8 (*CXCL8*), interleukin 1 beta (*IL1B*), *IL18*, nitric oxide synthase 2 (*NOS2*), nuclear factor kappa B subunit 1 (*NFKB1*), RELA proto-oncogene (*NFKB/p65*), toll-like receptor 4 (*TLR4*), toll-like receptor 5 (*TLR5*), myeloid differentiation primary response gene 88 (*MYD88*), transforming growth factor beta 1 (*TGFB1*), beta defensin 1 (*DEFB1*), beta defensin 2 (*DEFB4A*), and reference genes glyceraldehyde 3-phosphate dehydrogenase (*GADPH*) and hypoxanthine phosphoribosyltransferase 1 (*HPRT1*) were described in previous studies (Table 1). The relative normalized expression of the selected genes was assessed using the 2^−ΔΔCt^ method [68], comparing different conditions. Samples scored negatively when the Ct was ≥39.

### 2.4. Cytokine Quantification

The cytokine content was investigated in culture supernatants of IPEC-J2 described in Section 2.2.3, using both experimental designs. Cells were treated for 24 h with OMWW-EP (0.35 and 1.4 µg) without infection. Moreover, after OMWW pre-treatment, cells were infected (1 h) with *S. typhimurium* after a polyphenolic pre-treatment, alongside the corresponding controls. The culture medium was changed, and the cells were incubated for 3 h at 37 °C under 5% CO_2_. Then, culture supernatants were collected, centrifuged (at 2500× *g* for 3 min), and kept at -80 °C until use. Levels of GM-CSF, IL-1α, IL-1β, IL-1Ra, IL-6, IL-8, IL-10, IL-18 were determined using the Porcine Cytokine/Chemokine Magnetic Bead Panel Multiplex assay (Merck Millipore, Darmstadt, Germany) and a Bioplex MAGPIX Multiplex Reader (Bio-Rad, Hercules, CA, USA), following the manufacturer’s instructions [48].

### 2.5. Statistical Analyses

A Kolmogorov–Smirnov test was conducted to check for Gaussian distribution in the data sets, concerning the viability assay, gene expression, cells invasion, and protein release. Data showing Gaussian distributions were checked for significant differences by one-way ANOVA or unpaired T-test. Results failing the Kolmogorov–Smirnov test were checked for significant differences by non-parametric Kruskal–Wallis test, followed by a Dunn’s Multiple Comparison post-hoc test. The significance threshold was set at *p* < 0.05 (Prism 5, GraphPad Software, GraphPad Software Inc., San Diego, CA, USA).

## 3. Results

### 3.1. Cell Viability

Cells from the IPEC-J2 line cells were exposed to scalar doses of OMWW-EP (0.35, 0.7, 1.4, 7, 14, 70, 140 µg), and 24 h later, viability was measured through an XTT assay. The XTT viability test showed that treatment with OMWW-EP at 140 μg and 70 μg induced a statistically significant (*p* < 0.0001) decrease in IPEC-J2 viability after 24 h (Figure 1) OMWW-EP exposition. The other concentrations tested did not show a significant effect. The two doses (0.35, 1.4 µg) not affecting IPEC-J2 viability were therefore selected for the following experiments.

### 3.2. Salmonella Typhimurium Invasiveness

A significant (*p* < 0.05) decrease in *S. typhimurium* invasiveness into IPEC-J2 cells (*p* < 0.05; log_10_ CFU/3 × 10^5^ cells) after an exposure to OMWW-EP of 1.4 μg for 24 h was demonstrated when compared with controls (untreated infected cells). CFU data were converted into log_10_ values (Figure 2).

### 3.3. Modulation of Immune Response

The immunomodulant effect of OMWW-EP at two dosages (0.35 µg and 1.4 µg) was monitored through RT-qPCR and ELISA tests.

#### 3.3.1. OMWW-Extract Polyphenols’ Effect on IPEC-J2 Gene Expression and Cytokine Release

The effect of OMWW-EP (0.35 μg and 1.4 μg) treatment for 24 h on IPEC-J2 cells was monitored through RT-qPCR. A panel of seven genes was analyzed (Table 1), and the levels of treated cells were compared to untreated control cells. Moreover, complete results for the other polyphenol dosages are reported in the Appendix A. A significant decrease in *CXCL8* (*p* < 0.001), *IL18* (*p* < 0.05), and *MYD88* (*p* < 0.001) and a significant increase in *NOS2* (*p* < 0.05) were observed in cells exposed to 0.35 µg of OMWW-EP (Figure 3). The treatment with 1.4 µg of OMWW-EP triggered a significant decrease in *CXCL8* (*p* < 0.001) and *MYD88* (*p* < 0.001). The other genes under study were not significantly modulated (Figure 3).

#### 3.3.2. OMWW-Extract Polyphenols’ Effect on IPEC-J2 Cytokine Release

In parallel, the impact of scalar doses of OMWW-EP (0.35, 1.4 μg) on cytokine levels in IPEC-J2 culture supernatants was investigated using multiplex ELISA. Eight cytokines were tested: IL-1α, IL-1β, IL-1Ra, IL-6, IL-8, IL-10, IL-18 and GM-CSF (Figure 4). The levels of GM-CSF and IL-β were below the assay detection limit. Exposure to polyphenols did not alter the levels of IL-1α, IL-1Ra, and IL-10 in IPEC-J2 culture supernatants (Figure 4). In agreement with the RT-qPCR data, these compounds decreased the levels of the pro-inflammatory cytokines IL-6, IL-8, and IL-18, although a statistically significant difference was observed only for the latter (Figure 4).

#### 3.3.3. OMWW-Extract Polyphenols’ and *S. typhimurium* Infection Effects on IPEC-J2 Gene Expression

The effect of OMWW-EP pre-treatment for 24 h in IPEC-J2 responses to *S. typhimurium* infection (for 1 h) was evaluated. First, a panel of 10 genes was analyzed through RT-qPCR (Table 1). We observed a significant increase in *CXCL8* (*p* < 0.0001), *MYD88* (*p* < 0.0016), *DEFB1* (*p* < 0.0001), and *DEFB4A* (*p* < 0.0044) in cells not exposed to OMWW (control) in response to *S. typhimurium infection* (Figure 5).

Then we investigated the impact of OMWW-EP in IPEC-J2 ability to respond to *S. typhimurium* infection. Different results were obtained depending on polyphenol dosages. The pre-treatment of infected cells with 0.35 µg induced a significant decrease in *IL1B* (*p* = 0.019), *MYD88* (*p* = 0.062), *DEFB1* (*p* < 0.0001) and *DEFB4A* (*p* = 0.0012) compared to untreated infected cells, whereas with a pre-treatment of infected cells using 1.4 µg showed a significant decrease in *IL1B* (*p* = 0.019) and *DEFB4A* (*p* = 0.023) and a significant increase in *TGFB1* (*p* = 0.008) (Figure 5). Other genes under study were not significantly modulated (Figure 5).

#### 3.3.4. OMWW-Extract Polyphenols’ and *S. typhimurium* Infection Effects on IPEC-J2 Cytokine Production

To further investigate the immunomodulatory properties of OMWW-EP, we assayed the cytokine contents in the supernatants of un-infected and untreated IPEC-J2 and *S. typhimurium*-infected cells (ST) pretreated or not with OMWW-EP (0.35 μg or 1.4 μg) (Figure 6). *S. typhimurium* infection triggered an enhanced release of pro-inflammatory cytokines, such as IL-1α (*p* = 0.02927), IL-6 (*p* < 0.0001), and IL-8 (*p* = 0.0006) (Figure 6). OMWW-EP did not affect the cytokine content in *S. typhimurium*-infected IPEC supernatants (Figure 6). Values of GM-CSF and IL-1β were below the reference range values.

## 4. Discussion

Gut epithelial cells have a predominant role as the first defense from pathogenic insults [53,57]. Thus, the obtained results regarding the modulation of immune genes in the intestinal epithelium after treatment with polyphenols are a prodromal step to feed supplementation with polyphenols in livestock species. The IPEC-J2 cell line was chosen on the basis of our previous studies [48,49,50,52]. Indeed, it represents a good model for investigating epithelial immune response in pigs, in order to evaluate the ability of OMWW polyphenols to modulate the in vitro gut immunological response to *S. typhimurium* infection.

Our screening of different amounts of OMWW-extract polyphenols (OMWW-EP) carried out in the first part of the study through the IPEC-J2 viability test allowed us to choose the appropriate dosages for the successive analyses (Figure 1). We then investigated the effect of pre-treatment with OMWW-EP (0.35 and 1.4 μg) on IPEC-J2 cells for 24 h with or without an infective insult and found that exposure to these compounds triggered a decreased expression of *CXCL8*, *IL18*, and *MYD88* genes; IL-18 release; and an up-regulation of *NOS2* gene expression (Figure 3 and Figure 4). The anti-inflammatory effect of polyphenols is due to complex cellular mechanisms that are still not clear, but most of them have been correlated with the NF-kB pathway [45]. *CXCL8* is a pro-inflammatory chemokine (IL-8)-encoding gene whose expression could be regulated by the TLR4/MyD88/NF-kB pathway. In our study, OMWW-EP seemed to exert anti-inflammatory action by decreasing *MYD88* (a gene with a pivotal role in NF-kB activation) and therefore *CXCL8* gene expression. Indeed, in other studies, dietary supplementation with grape seed cake (another by-product rich in polyphenols) was shown to significantly reduce *MYD88* gene expression in the colon of Dextran Sulfate Sodium (DSS)-treated piglets [70], and Li et al. [71] demonstrated the ability of other natural polyphenols (the flavonoids quercetin and catechin) to restore the increased expression of *MYD88* in LPS-stimulated murine macrophage RAW 264.7 cells.

In addition, our data demonstrated the ability of OMWW-EP to decrease both the expression and secretion of IL-18. IL-18 is a member of the IL-1 family, with an important role in the inflammatory response [72,73,74,75,76,77]. Its release must be tightly controlled, in order to avoid the development of auto-inflammatory diseases [73].

These data suggest a possible effect of OMWW polyphenols on host–pathogens interaction, which was successively tested *in vitro* using *S. typhimurium* assay. In this way, the ability of a pre-treatment with different dosages of OMWW-EP (0.35 and 1.4 µg) to decrease *S. typhimurium* invasiveness and modulate immune response related-genes in *S. typhimurium*-infected cells was assessed. First of all, our data confirmed *S. typhimurium’s* ability to penetrate IPEC-J2 cells (Figure 2), which is known to be related to the up-regulation of the pro-inflammatory molecule *CXCL8* [49], as we found (Figure 5 and Figure 6). In our analysis, we additionally found that *S. typhimurium’s* invasion of IPEC-J2 significantly increases the expression of *MYD88* gene (Figure 5) encoding for the MyD88 adaptor protein, which is the mediator of NF-kB activation, essential for the stimulation of pro-inflammatory gene expressions. Not surprisingly, we therefore observed the increased release of other pro-inflammatory cytokines (IL-1α, IL-6, IL-18) after *S. typhimurium* infection, in accordance with a previous study [49] (Figure 6). It is known that mucosal bacteria are able to stimulate the transcription of pro-inflammatory genes through epithelial cell invasion, interacting with many receptors such as TLR or acting on NF-kB [44]. Surface-expressed TLRs are activated by the pathogen-associated molecular patterns (PAMPs), which are microbe structures, exploiting the adaptor molecule MyD88 and stimulating NF-kB translocation into the nucleus [49]. The activation of the nuclear factor NF-kB leads to the increased transcription of pro-inflammatory mediators (such as the cytokines IL-8, IL-1B, IL-6 and TNF) [78], as shown in our experiment (Figure 6).

Interestingly, the pre-treatment with OMWW extract enriched in polyphenols reduced *S. typhimurium* invasiveness. Thus, we tried to highlight molecules influencing this host–pathogen interaction, modulated by the polyphenol treatment. Firstly, the expression of two TLRs (*TLR4* and *TLR5*) was investigated, but no effects on *S. typhimurium* invasion after OMWW-EP pre-treatment were observed.

As for the effects on pro-inflammatory cytokines and related pathways, we observed a down-regulation of *IL1B* in *S. typhimurium*-infected IPEC-J2 cells after OMWW-EP pre-treatment (both dosages). A down-regulation of *MYD88* for the 0.35 µg group was detected as well.

The expression of *MYD88* was also investigated and showed a decrease in *S. typhimurium*-infected cells in the 0.35 µg pre-treatment group, while *MYD88* gene expression was raised in cells infected and not pre-treated with OMWW-EP. The pre-treatment with 0.35 µg OMWW-EP probably prevented the activation of NF-kB and pro-inflammatory mediators through the down-regulation of this adaptor molecule-encoding gene (*MYD88*).

Moreover, inflammasome-induced cell death contributes to host control of *S. typhimurium* infection. Species differences in inflammasomes may contribute to zoonotic immune tolerance. Inflammasomes are molecular platforms that promote the maturation of the proinflammatory cytokines IL-1β and IL-18. During enteric *Salmonella* infection, the activation of caspase-1 and the production of IL-1β and IL-18 provide a protective host response [79]. The inflammasome activation could be mediated by MyD88, but there are other pathways in the activation signaling: various PAMPs, DAMPs, or intracellular changes induce the formation of the NLRP3 inflammasome composed of NLRP3 as a PRR, pro-caspase-1, and adapter proteins such as the apoptosis-associated speck-like protein containing a caspase recruitment domain [80].

At both dosages (0.35 and 1.4 µg), the polyphenolic pre-treatment induced a down-regulation of pro-inflammatory cytokine *IL1B*, which is also involved in the inflammasome reaction, together with IL-18 [73,74,75,76,77]. The combined effect of OMWW-EP on IL-18, which induced gene expression and reduced cytokine release (without *Salmonella* infection), leads the authors to suppose that this is the pathway through which OMWW-EP potentially protects IPEC-J2 cells against *S. typhimurium* infection.

In line with our results, the ability of the polyphenol resveratrol to potentially protect the intestinal barrier against deoxynivalenol (DON)-induced dysfunction and *Escherichia coli* (*E. coli*) translocation in IPEC-J2 cells [64] and *S. typhimurium* infection was demonstrated. Several *in vitro* studies concerning intestinal cells demonstrated that plant extracts rich in polyphenols or isolated molecules can limit induced-inflammation processes [30,81,82,83,84]. It was also shown that natural polyphenols can modulate inflammasome activation [77], interfering with the production (both at mRNA and protein levels) of pro-inflammatory mediators [30]. Moreover, it has been demonstrated that IL1-β is reduced by polyphenols such as curcumin and resveratrol [84,85,86]. Other good sources of polyphenols, i.e., dietary grape seed cake, decreased *IL1B* gene expression and protein concentration in fattening pigs’ spleens [87]. Feeding weaned pigs with polyphenol-rich plant products (grape seed, grape marc meal extract, and spent hops) down-regulated various pro-inflammatory cytokines, including IL-1β [44], in the intestine, and the oleuropein glycoside polyphenol significantly decreased the release of IL1-β in LPS-stimulated human whole-blood cultures [88].

Meanwhile, we did not observe differences between OMWW-EP-treated and untreated IPEC-J2 cells concerning the expression and release of other pro-inflammatory cytokines in response to *S. typhimurium* infection. This probably relates to the fact that these inflammatory molecules are primarily stimulated by TLR4 receptors, whose expression seemed to not be significantly modulated by polyphenols. We also might speculate that OMWW-EP could reduce the levels of pro-inflammatory cytokine release in response to *Salmonella* if a lower infective dose is used.

Not only pro-inflammatory but also anti-inflammatory cytokines such as IL-10 and TGF-β were tested. In particular, TGF-β can dampen the inflammatory effects of cytokines such as IL-1β, IL-12, TNF [89].

In our experiments, we observed that *S. typhimurium* infection determined a decrease in *TGFB1* expression in IPEC-J2 cells. This is not surprising, considering a recent study by Qin et al. [90], who mimed the bacterial infection process with an LPS stimulus in human Caco-2 colon cells. The authors observed a down-regulation of several genes involved in the inflammation response linked to TGF-beta signaling pathways. Interestingly, the pre-treatment with OMWW-EP (1.4 µg) induced the up-regulation of *TGFB1* in infected cells compared to cells that were infected without OMWW-EP pre-treatment. The *TGFB1* gene encodes the TGF-beta superfamily ligands and binds different TGF-beta receptors, regulating gene expression as well as cell growth, proliferation, and differentiation [91]. It is produced by different cell types, including the intestinal cells [92], and is a cytokine involved in the homeostasis of the epithelial barrier, which is normalized by up-regulating the expression of tight junction proteins [93]. The up-regulation of this gene in our experiments may be correlated with a possible inhibitory action of bacterial replication inside the cells, in line with Huang et al. [91]., who demonstrated that, in pigs, the inhibition of *S. typhimurium* intracellular replication can be associated with *TGFB1* increase. Moreover, in a study by Nallathambi et al. [93], a polyphenol-rich grape seed extract was able to enhance *TGFB1* expression in Caco-2 cells, in line with the observed increase in tight junction protein expression.

Finally, the expression of genes coding for antimicrobial peptides (AMPs), released during early response to invading pathogens, was investigated. These molecules show efficacy in disrupting both the Gram-positive and Gram-negative bacterial membranes and are also expressed in epithelial cells of the gastrointestinal tract [26]. Beta-defensins are known AMPs also involved in the maintenance of the homeostasis in the gut microbiota, regulating its composition and thus protecting from microbial pathogens [48,75,94], and it is well known that IPEC-J2 cells express beta-defensin genes [26]. Our results showed that *S. typhimurium* invasion up-regulated *DEFB1* and *DEFB4A* gene expressions in IPEC-J2 cells, as reported by previous studies [26,49]. *DEFB4A* expression was also increased in another porcine ileum epithelial cell line (IPI-2I) after infection with *S. typhimurium* DT104 [26]. Furthermore, it was demonstrated that *E. coli* adhesion increases the expression of *DEFB1* and *DEFB4A* in IPEC-J2 cells [95]. In this study, pre-treatment with OMWW-EP seemed to induce a return to the basal expression of *DEFB1* (at the 0.35 µg dosage) and *DEFB4A* (at both dosages) after *S. typhimurium* infection, counteracting the effect of bacterial invasion and potentially restoring gut homeostasis.

## 5. Conclusions

Our results confirmed the potential ability of OMWW-EP to modulate host–pathogen interactions in pigs by inducing an alteration of *S. typhimurium* invasiveness. In particular, our data showed a significant reduction of *S. typhimurium’s* ability to invade cells pre-treated with 1.4 µg of OMWW-EP compared to untreated IPEC-J2 cells. Furthermore, pre-treatment (independently from the dosage) with OMWW-EP modulated several innate immune-response genes influencing the *S. typhimurium* invasiveness in IPEC-J2, exhibiting potential antimicrobial activity by decreasing intracellular bacterial replication. This is the first study performed in an *in vitro* swine intestinal model that suggests a potential protective role of OMWW polyphenols in the pig intestine, paving the way for *in vivo* studies to confirm these promising results; increasing our knowledge of related molecular mechanisms; and making the possible use of this by-product feasible for livestock animal welfare and health. 

## Figures and Tables

**Figure 1 animals-14-00564-f001:**
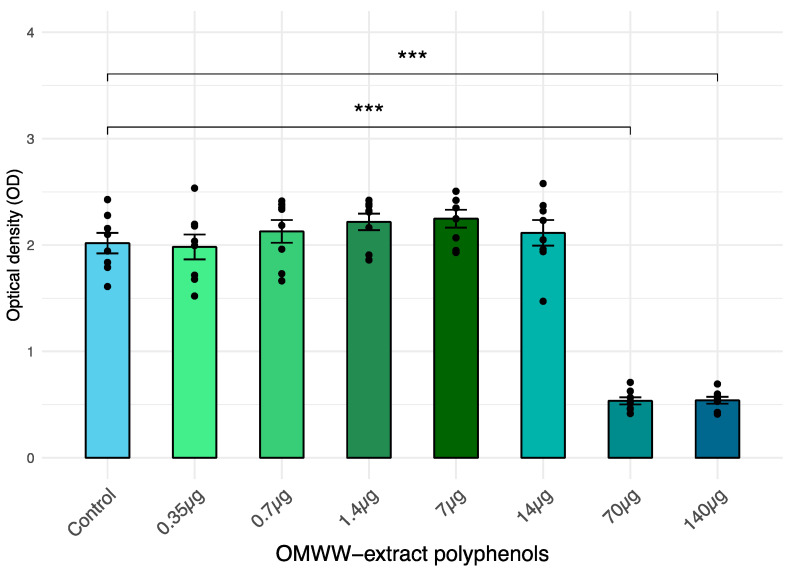
IPEC-J2 viability after a 24 h exposition to the extract of liquid olive mill waste-water (OMWW) polyphenols (OMWW-EP: 0.35 μg; 0.7 μg; 1.4 μg; 7 µg; 14 µg; 70 µg; 140 µg). Cell viability was determined with XTT test. The number of living cells is expressed as optical density (OD) ± standard error (SE) and dots indicate samples included in each group. Statistical difference was calculated for all groups vs. Control (untreated cells): *** *p* < 0.001.

**Figure 2 animals-14-00564-f002:**
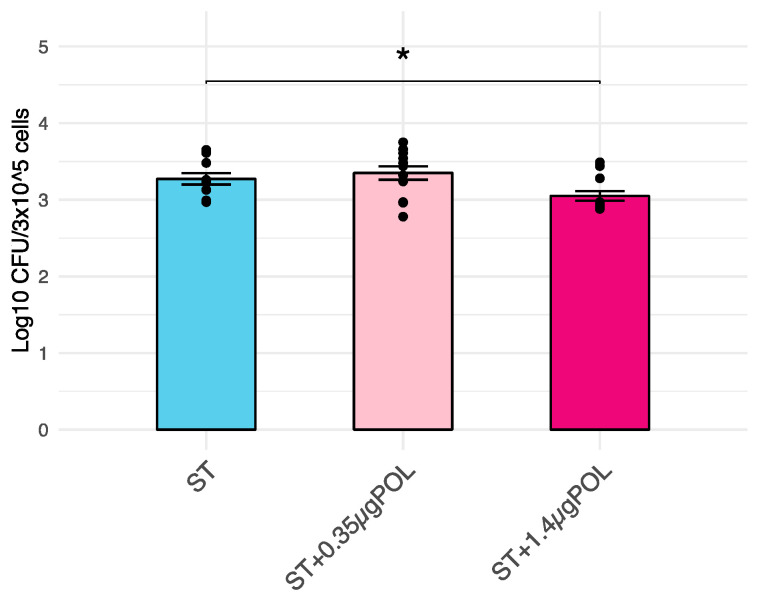
Effects of OMWW-extract polyphenols (OMWW-EP) on *S. typhimurium* penetration into IPEC-J2 cells. Data are expressed as log_10_ CFU of penetrated, intracellular ST/3 × 10^5^ cells. The mean value of five replicates + standard error is presented, and dots indicate samples included in each group. The significant difference between *S. typhimurium* infected cells and pretreated with different concentrations of OMWW-EP (ST + 0.35 µg POL—pink; ST + 1.4 µg POL—fuchsia) and *S. typhimurium* infected cells (ST—blue) is indicated by * (* *p* < 0.05).

**Figure 3 animals-14-00564-f003:**
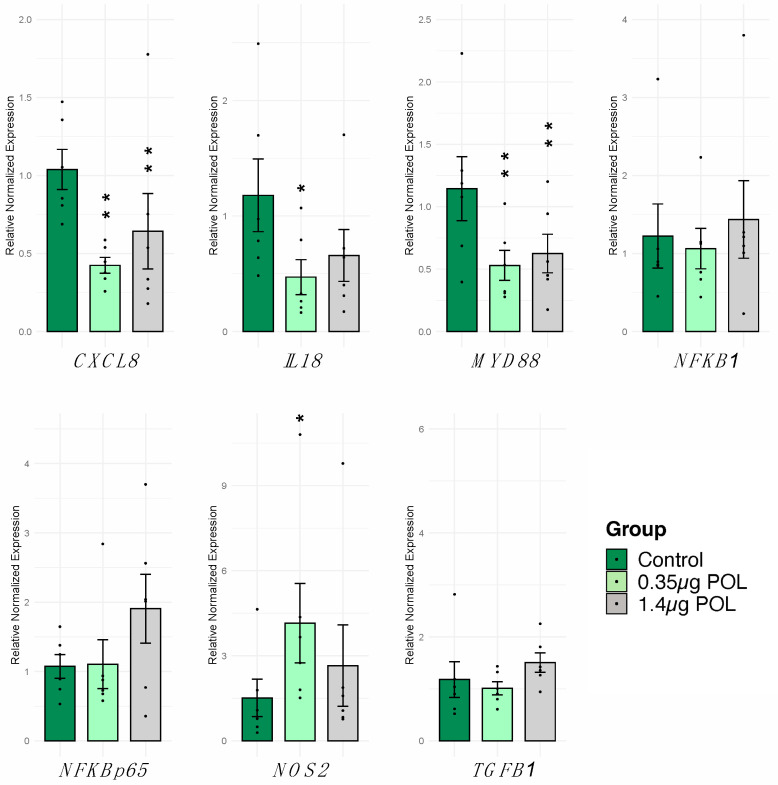
Effects of 24 h OMWW-extract polyphenols on IPEC-J2 gene expression. The RT-qPCR analysis was performed to evaluate *CXCL8*, *IL18*, *MYD88*, *NFKB1*, *NFKB/p65*, *TGFB1* and *NOS2* gene expression. Data are presented as bar plots displaying the mean value of normalized expression, standard error as error bars and dots indicate samples included in each group. For each gene and cytokine, differences between treated with polyphenols (0.35 μg POL—light green; 1.4 μg POL—gray) vs. untreated (Control—dark green) cells were evaluated through one-way ANOVA followed by a Dunnett’s test or a Kruskal–Wallis test followed by Dunn’s multiple comparison test; * *p* < 0.05, ** *p* < 0.01.

**Figure 4 animals-14-00564-f004:**
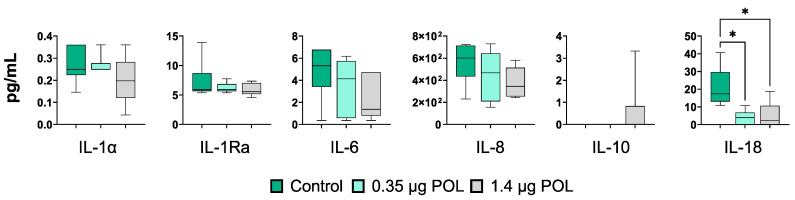
Effects of OMWW-extract polyphenols on IPEC-J2 cytokine release through multiplex ELISA measuring cytokine contents in culture supernatants (IL-1α, IL-1Ra, IL-6, IL-8, IL-10, IL-18). Data are presented as box and whisker plots displaying median and interquartile range (boxes) and minimum and maximum values (whiskers). For each gene and cytokine, differences between treated with polyphenols (0.35 μg POL—light green; 1.4 μg POL—gray) and untreated (Control—dark green) cells were evaluated through one-way ANOVA followed by a Dunnett’s test or a Kruskal–Wallis test followed by Dunn’s multiple comparison test; * *p* < 0.05.

**Figure 5 animals-14-00564-f005:**
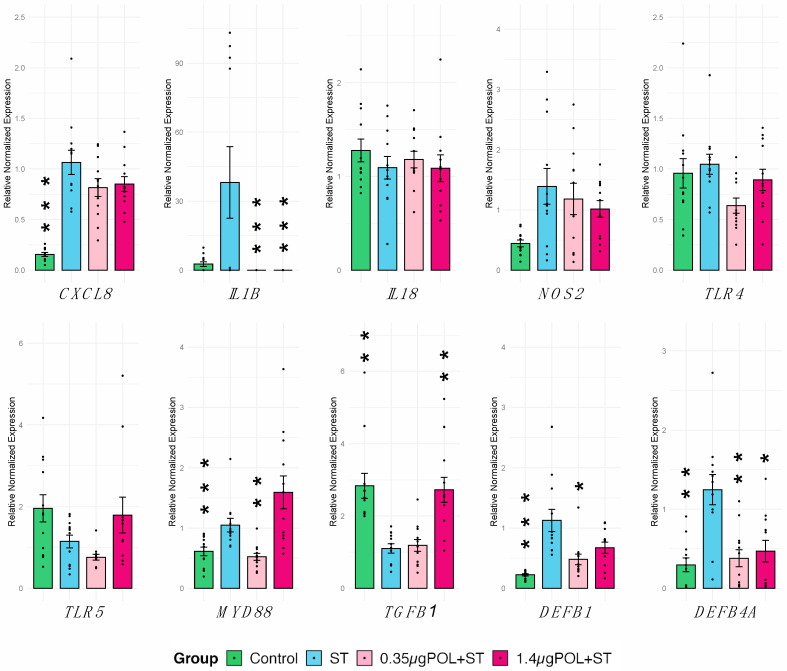
Gene expression of IPEC-J2 cells in response to *S. typhimurium* infection with OMWW-extract polyphenol (OMWW-EP) pre-treatment. The tested conditions for IPEC-J2 cells were: uninfected and untreated cells (Control—dark green), infected with *S. typhimurium* (ST—blue), pre-treated with 0.35 µg OMWW-EP and infected (0.35 µg POL + ST—pink), and pre-treated with 1.4 µg OMWW-EP and infected (1.5 µg POL + ST—fuchsia). Data are reported as mean value and standard error, and dots indicate samples included in each group. Statistical tests were carried out comparing all conditions vs. ST. Significant differences: * *p* < 0.05, ** *p* < 0.01, *** *p* <0.001.

**Figure 6 animals-14-00564-f006:**
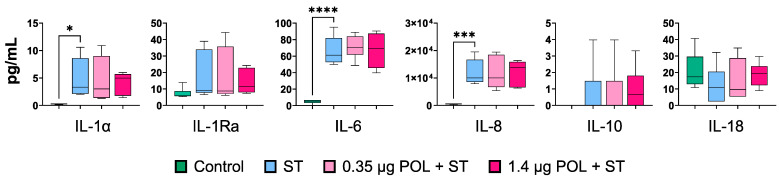
Cytokine release by IPEC-J2 cells in response to *S. typhimurium* infection after OMWW-extract polyphenol (OMWW-EP) pre-treatment. The tested conditions for IPEC-J2 cells were: uninfected and untreated cells (Control—dark green), infected *with S. typhimurium* (ST—blue), pre-treated with 0.35 µg OMWW-EP and infected (0.35 µg POL + ST—pink), and pre-treated with 1.4 µg OMWW-EP and infected (1.5 µg POL + ST—fuchsia). For each cytokine, differences between ST-infected cells and the other conditions were evaluated through one-way ANOVA followed by a Dunnett’s test or a Kruskal–Wallis test followed by Dunn’s multiple comparison test; * *p* < 0.05, *** *p* < 0.001; **** *p* < 0.0001.

**Table 1 animals-14-00564-t001:** Primer set sequences of target and reference genes.

**Gene**	**Primer Sequences**	**Amplicon Length**	**Source**
*IL18*	For-5′-CGTGTTTGAGGATATGCCTGATT-3′ Rev-5′-TGGTTACTGCCAGACCTCTAGTGA-3′	106	[48]
*IL1B*	For-5′-AATTCGAGTCTGCCCTGTACCC-3′ Rev-5′-TGGTGAAGTCGGTTATATCTTGGC-3′	110	[49]
*NOS2*	For-5′-CGTTATGCCACCAACAATGG-3′ Rev-5′-AGACCCGGAAGTCGTGCTT-3′	84	[48]
*TGFB1*	For-5′-CGCGTGCTAATGGTGGAAAG-3′ Rev-5′-CCGACGTGTTGAACAGCATA-3′	87	[48]
*CXCL8*	For-5′-TTCGATGCCAGTGCATAAATA-3′ Rev-5′-CTGTACAACCTTCTGCACCCA-3′	175	[69]
*MYD88*	For-5′-GCAGCTGGAACAGACCAACT-3′ Rev-5′-GTGCCAGGCAGGACATCT-3′	62	[69]
*NFKB1*	For-5′-CCCATGTAGACAGCACCACCTATGAT-3′ Rev-5′-ACAGAGGCTCAAAGTTCTCCACCA-3′	131	[69]
*NFKB/p65*	For-5′-CGAGAGGAGCACGGATACCA-3′ Rev-5′-GCCCCGTGTAGCCATTGA-3′	61	[69]
*DEFB1*	For-5′-CTGTTAGCTGCTTAAGGAATAAAGGC-3′ Rev-5′-TGCCACAGGTGCCGATCT-3′	80	[48]
*DEFB4A*	For-5′-CCAGAGGTCCGACCACTA-3′ Rev-5′-GGTCCCTTCAATCCTGTT-3′	87	[48]
*TLR4*	For-5′-TGGCAGTTTCTGAGGAGTCATG-3′ Rev.–5′ –CCGCAGCAGGGACTTCTC-3′	71	[48]
*TLR5*	For-5′-TCAAAGATCCTGACCATCACA-3′ Rev.-5′ –CCAGCTGTATCAGGGAGCTT-3′	59	[48]
*GAPDH*	For-5′-ATGGTGAAGGTCGGAGTGAA-3′ Rev-5′AGTGGAGGTCAATGAAGGGG-3′	61	[48]
*HPRT1*	For-5′-AACCTTGCTTTCCTTGGTCA-3′ Rev-5′-TCAAGGGCATAGCCTACCAC-3′	150	[48]

## Data Availability

The data presented in this study are available in article and Appendix A.

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
