# Peer review of "Olive Mill Waste-Water Extract Enriched in Hydroxytyrosol and Tyrosol Modulates Host–Pathogen Interaction in IPEC-J2 Cells"

_animals, 2024, doi:10.3390/ani14040564_

Round 1
Reviewer 1 Report
Comments and Suggestions for Authors
The main change request concerns the materials and methods. Specifically, the number of plates / replicates / wells for each replicate carried out in each part of the experiment is not clear (for example in lines 148-149). Even in paragraph 2.2.2 it is not clear for each experiment how many replicates and wells per replicate were used for both the treatments and the control.
In discussions, the description of all experiments that have already been previously explained in the materials and methods could be shortened.
Comments on the Quality of English LanguageA revision of the English would be appropriate since in some parts of the introduction and discussions the meaning of the speech is difficult to understand or there are some grammatical errors (for example in line 54, 106-107, 168, 340-341, 378-379 , 405-406)
Reviewer 2 Report
Comments and Suggestions for Authors
Comments for the Author:
The study by Ferlisi F et al, titled “Olive Mill Waste Water (OMWW) polyphenols modulate host-pathogen interaction in IPEC-J2 cells”. It falls completely short of the requirements for the publications.
Author showed that OMWW polyphenols modulates basal expression of inflammatory genes (Fig 3) and IL-18 secretion (Fig 4). However, these immunomodulatory effects were observed only at RNA level not at protein level in S. typhimurium treated cells. How OMWW polyphenol is useful??
Reviewer 3 Report
Comments and Suggestions for Authors
This manuscript investigated the effects of OMWW polyphenols on the immune response of porcine epithelial cells (IPEC-J2) following Salmonella typhimurium infection. The study focuses on an innovative application of OMWW polyphenols, a by-product of olive oil production, in animal diet, potentially reducing the use of antibiotics in livestock and the risk of antimicrobial resistance. The study employs robust experimental methods, including the use of a well-established in vitro model (IPEC-J2 cell line), various doses of OMWW polyphenols, and a range of assays (e.g., RT-qPCR, ELISA tests) to evaluate bacterial invasiveness and gene expression changes. The results indicate that OMWW polyphenols can significantly reduce S. typhimurium invasiveness and modulate immune gene expression in IPEC-J2 cells, suggesting potential immunomodulatory and antimicrobial activities.
The study is conducted entirely in vitro, and while the findings are promising, they need validation in live animals to establish real-world applicability in the future.
The study focuses mainly on two concentrations of OMWW polyphenols. Exploring a broader range of concentrations could provide more comprehensive insights.
Table 1: The header in the middle of Table 1 can be deleted.
Figures: How many samples for each group? Please use dots to represent the values of each sample in the bars. The title above the plots can be removed.
The discussion is too long and repeated the test results too much. It is suggested to simplify and focus on the significance of the results of each index and the comparison with other similar studies.
In conclusion, this manuscript presents valuable insights into the potential use of OMWW polyphenols in modulating host-pathogen interactions in porcine epithelial cells. Its findings contribute to the understanding of alternative approaches to managing intestinal health in livestock. However, further research, particularly in vivo studies, would be beneficial to fully ascertain the practical implications of these findings.
Round 2
Reviewer 1 Report
Comments and Suggestions for Authors
There are no further comments
Reviewer 2 Report
Comments and Suggestions for Authors
Author mentioned that “this experiment helps to confirm that polyphenols could contribute, as adjuvantor in preventive approaches, to the treatment of chronic inflammatory diseases”. However, the available evidence does not support the hypothesis.